# Development of a Nutraceutical Product Derived from By-Products of the Lipid Extraction of the Brazil Nut (*Bertolletia excelsa* H.B.K)

**DOI:** 10.3390/foods12071446

**Published:** 2023-03-29

**Authors:** Orquidea Vasconcelos dos Santos, Glaucinéia Oliveira Azevedo, Ângela Chagas Santos, Alessandra Santos Lopes

**Affiliations:** 1Graduate Program in Food Science and Technology (PPGCTA), Federal University of Para (UFPA), Belém 66075-110, PA, Brazil; 2Evandro Chagas Institute, Ananindeua 67030-000, PA, Brazil

**Keywords:** protein, by-products, nutraceutical, bars

## Abstract

The objective of this investigation was to develop a nutraceutical product, in bars, using defatted Brazil nut by-products. The methodological bases of analyses followed the guidelines of the Association of Official Analytical Chemists international and experimental tests of formulations. The bars presented with high protein content with high nutraceutical potential expressed as high levels of magnesium, sodium, calcium and potassium, with emphasis on selenium, supplying around 50% of the daily recommendations. The sulfur amino acids reach approximately the recommended chemical score (97%), the aromatic amino acids with a proportion close to 50% and those of the branched chain supplying the established recommendations for adults. In the sensory tests, the formulation that presented the best attributes and the greatest acceptance (91.78% for the flavor and 97.89% for the general impression) by the tasters was the bar formulation made with a 75:25 ratio of defatted Brazil nuts flour and isolated soy protein.

## 1. Introduction

The great importance of Amazonian fruits in the Brazilian trade balance is due primarily to oilseed, specifically Brazil nuts (*Bertolletia excelsa* H.B.K), that have a high rate of internal and external consumption of whole almonds coated and uncoated. However, a considerable amount of Brazil nuts does not meet top standard, being sold broken and still with peel, having less commercial value, or being used for the extraction of the lipid content the post-extraction residue of this almond is treated as industrial waste, which bears a considerable impact to the environment. 

Research has shown that industrial Brazil nuts by-products have high nutritional importance, due to high levels of nutritional and functional constituents in these raw materials. These constituents are referred to as functional foods, more specifically nutraceuticals that feature substances considered foods and part of them providing beneficial health effects, either as a preventive or supporting element in the treatment of pathologies. According to data from GlobeNewswire report [1], the world growth of this economic sector is estimated to be 7% between 2016 and 2022, with an investment perspective of around 300 billion Euros until mid-2022.

It is estimated that the production of industrial by-products with nutraceutical characteristics will reach approximately 1.3 billion tons per year and that over 50% of this amount is derived from fruit crops. Research with these materials has shown extensive possibilities of increasing the production chain of this segment, with the reuse and expansion of a chain of products with new designs elaborated based on the inclusion of by-products as strategies for the development of new industrial segments, generation of new market franchises, increased profitability, adding value and in parallel reducing environmental waste [2]. 

The insertion of these nuts in the segment of products with high practicality as in bars, enhances the profile of consumers seeking daily food based on the tripod nutrition, prevention and consumption convenience [3]. The snacks market segment, where the bars are inserted, has a global growth projection of US$ 256 million until 2026 driven by the increase in the vegan population, the increase in the fitness market and the search for healthy food sources [1,4].

In view of the source of nutritional and functional components of this oilseed and the low commercial investment in broken and peeled nut kernels, combined with the market expectations in the snack bar segment, this investigation aims to develop a nutraceutical product, like bars using the defatted by-product of Brazil nut (*Bertolletia excelsa* H.B.K).

## 2. Material and Methods

### 2.1. Raw Material

About 5 kg of Brazil nuts were used, from the company Benedito Mutran Cia Ltd. (Belém, PA, Brazil) a, broken and depeliculated, packed in flexible laminated packaging and cardboard box. The samples were transported and stored in styrofoam boxes at the Food Science Laboratory at the Federal University of Pará at a temperature of 7 °C. The other ingredients used: soy protein isolate (SPI), brand Supro^®^ 783 (The Solae Company, São Paulo, SP, Brazil) with 90% protein; invert sugar (Berasugar 55) and glucose syrup (Beraca Sabara, São Paulo, SP, Brazil), obtained commercially-Sao Paulo BR.

### 2.2. Methods

#### 2.2.1. Microbiological and Mycotoxicological Analyzes on Brazil Nuts

The samples were submitted to microbiological analysis according to tolerance parameters required by Resolution RDC n° 12, of 2 January 2001 [5] according to methodologies described by Vanderzant and Splittstoesser [6], for coliforms at 45 °C and *Salmonella* sp. and aflatoxins B1, B2, G1 and G2, according to the thin layer chromatography method described by Soares and Rodriguez-Amaya [7], at the Laboratory of Food and Nutrition Analysis (LAN), of the School of Agriculture “Luiz de Queiroz”.).

#### 2.2.2. Process of Obtaining of the Defatted Flour Brazil Nut

The Brazil nut samples were previously submitted to the drying process at a temperature of 50 °C for 24 h in an oven with forced air circulation (Thoth model 510.150). The extraction of fat was carried out by hydraulic pressing (MARCON MPH-15) with a capacity of 15 tons of pressure. The pie was defatted ground in mill type Willye brand (TECNAL model TE650) producing the Defatted Flour of Brazil Nuts (DFBN).

#### 2.2.3. Determination of the Composition of Macro and Micronutrients of the Defatted Flour of Brazil Nuts

The analyzes were based on physicochemical analyzes of nutritional composition including Water activity: DECAGON’s AquaLab Series 3TE instrument; pH: according to the Association of Official Agricultural Chemists (AOAC) method [8]; Humidity: according to AOAC method no. 934.06 [8]; Crude protein micro Kjeldahl method no 920,152 from AOAC [8]; Total lipids method no 983.23 of AOAC [8]; Fixed mineral residue with AOAC method 940.26 [8]; Total fibers 985.29 enzymatic-gravimetric method AOAC [8]; Total carbohydrates and total energy value: calculated by difference and application of Atwater factors 4–9–4 kcal/g for proteins, lipids and total carbohydrates, respectively; according to RDC Resolution No. 429 of 9 October 2022 [9].

The samples were prepared (open) in duplicate with triplicate readings by acid digestion (closed system) using a Multiwave 3000 digestion system (Anton Paar, Graz, STM, Austria). The reading of the elements was performed by argon plasma atomic emission spectrometry (ICP-AES) coupled to a Spectro Smart Analyzer Vision Software (Genesis SOP, Ametek, Berwyn, PA, USA). 

#### 2.2.4. Formulation of the Bars

The formulations were elaborated from preliminary laboratory experiments. The best proportions of Brazil nut flour and soy protein isolate were determined, considering the following aspects: homogeneity of the dough, texture and especially the protein content. Three formulations (F1, F2 and F3) were subjected to macro and micronutrient evaluations and sensory acceptance tests.

The contents of Brazil nut flour and soy protein isolate add up to 50% of the dry ingredients content of the formulation. The agglutinating syrup was formed with the liquid components used: inverted sugar, glucose syrup and hydrogenated vegetable fat, making up the remaining 50% of the final formulation, as shown in Table 1.

The formulations were made from the mixture of these ingredients and their respective concentrations, transferred to flat surfaces, unrolled and cut into bars, measuring approximately 10 × 2 × 1 cm^3^, weighing 30 g each. The bars were individually wrapped in flexible film (PVC Strtch Film 280 mm × 300 mm), vacuum packed, covered with aluminum foil and stored at room temperature (≈25 °C) (Figure 1).

#### 2.2.5. Microbiological and Physicochemical Analysis of the Defatted Flour of Brazil Nuts and Bar Formulations

Microbiological analyzes were performed, with methodology similar to item Microbiological and Mycotoxicological Brazil Nuts Analysis and the physical chemistry of DFBN and the formulations of the Brazil nut bars evaluated, according to the analysis of the bar.

#### 2.2.6. Amino Acid Analysis in Bar Formulations Containing Defatted Flour of Brazil Nuts and Soy Protein Isolate

The quantification of amino acids present in the bars was performed at the Laboratory of Protein Sources of the Department of Food Planning and Nutrition, Faculty of Food Engineering. This quantification was performed by hydrolysis with 6 N hydrochloric acid for 24 h. The amino acids released during the acid hydrolysis react with phenyl isothiocyanate (PITC) and were separated in a High-performance liquid chromatography-(HPLC) (A53000—Thermo Separation Products) using a reversed phase HPLC C-18 column (Phenomenex, Macclesfield, Ches, England) and detected by UV absorptivity at 254 nm. Their quantification was performed using internal multilevel calibration and acid alpha-aminobutyric acid (creatine AAB) as the internal standard. The results obtained were assessed by calculating the chemical score (CS) of each amino acid (Equation (1)) and were compared with the Food and Agriculture Organization of the United Nations/World Health Organization [10] standards.
(1)$$CS=\frac{mg\;of\;amino\;acids/g\;of\;protein}{FAO-WHO\;standard}$$

#### 2.2.7. Sensory Analysis of the Formulated Bars

The three selected formulations of bars with a high protein content were evaluated sensorially by applying the acceptance tests with an unstructured hedonic scale, having in their extremes the terms I accepted very much (9) and did not accept very much (1). The attributes evaluated for the bars were: color, aroma, texture, taste, global impression and purchase intent test, conducted immediately after the acceptance test, being made by the same tasters, according to the methodology described by Stone and Sidel [11].

The team of 100 untrained tasters was recruited among students, employees and professors at the Federal University of Pará in the city of Belém-PA. The project was approved by the Standing Committee on Ethics in Research with Human Beings of the Federal University of Pará (CAAE nº 42176020.3.0000.0018). A free and informed consent term was previously applied, stating if they were habitual consumers of Brazil nuts and soy-based products and if they had any allergies to products or derivatives (taken as exclusion items).

The samples were presented monadic and casualizada manner, disposable plates coded with three-digit numerals. The acceptance index (AI) and purchase intention (PI) were obtained using Equation (2), where M is the average of the grades obtained and X the maximum score (9) for the acceptability test and 5 for the buy intention.
(2)$$AI\;or\;PI=\frac{M}{X}\times 100\%$$

#### 2.2.8. Statistical Analysis of the Results

The physical-chemical analyzes were performed in triplicate (mean ± standard deviation) and the results of the sensory tests obtained were subjected to statistical analysis, with the aid of the Software Statistica version 7.0 (Dell, Austin, TX, USA) [12] using the following statistical methodologies: The results of the physical-chemical and sensorial analyzes were submitted to analysis of variance (ANOVA) and the averages were compared by Tukey’s test with a significance of 5% of probability, with the aid of the statistical software. The ANOVA F values were compared with tabulated F values (FT), provided by Dutcosky [13]. The value of F greater than the value of FT indicates difference for the attribute, at the level of equivalent significance.

## 3. Results and Discussion

The different bar formulas showed average yields above or close to 98%; this value was calculated directly as the weight ratio of the ingredients and the weight of the final mass as expressed in Table 1.

### 3.1. Microbiological, Mycotoxicological and Physical-Chemical Analysis

The results of the microbiological and mycotoxicological analyses of fresh Brazil nut, DFBN and the formulations of the Brazil nut bars shown tha the number of coliforms at 45 °C (<3.0 MPN g^−1^) indicates that Brazil nut does not present a level of contamination that offers risks to consumers’ health; it also indicates that the preparation of flour and bar formulations were handled under good hygiene conditions, following the procedures of good formulation practices; the products were well below the maximum tolerance levels set by the legislation, with a maximum value of 10^2^ g^−1^ [5]. The absence of Salmonella sp indicates that both the handling of the raw material and its processing were efficient, that is, the product is fit for consumption [5]. The results of mycotoxin analyses, in the fractions B1, B2, G1 and G2, with a detection limit of 0.5 µg kg^−1^ showed that the fractions B1 and B2 had values below the maximum tolerated by the Brazilian legislation [14]. The type G1 and G2 aflatoxins were not detected considering this range limit. 

According to the Brazilian legislation [14] when the four types of aflatoxin are present, the maximum tolerance allowed for safe human consumption is the sum of the fractions (B1 + B2 + G1 + G2) with maximum total values of 20 µg kg^−1^. 

In this investigation, the microbiological and mycotoxicological evaluations aimed at proving the quality of the fresh raw material in its application in formulations and finished products according to the food safety standards established by the Ministry of Health in the Resolution of the National Health Surveillance Agency (ANVISA), RDC No. 274 [15].

The results of the physical-chemical analyses of the Defatted Flour of Brazil Nuts and of the Brazil nut bars formulas are shown in Table 2. 

The total energy value (TEV) found for DFBN flour was 402.7 kcal 100 g^−1^. When compared to sapucaia kernels flour [16], DFBN flour presented a lower energy value (447.74 vs. 665.22 kcal 100g^−1^). Among the macronutrients protein stands out most, similar to the protein content of sapucaia flours reported by Teixeira et al. [16], with an average of 20.45% to 49.28%, and in germinated (37.2%) and non-germinated (42%) soya bean flours [17], and in peanut flours of two varieties revealing protein averages from 26.17% to 27.42% [18].

The lipid content of Defatted Flour of Brazil Nuts remained on average, similar to that of sapucaia flour [16] with average levels from 18.33 to 58.71% and in germinated soya bean (15.2%) and not germinated soya flour (21%) [17]. These data vary, among other aspects, according to the sources and processes of lipid extraction used in these studies. 

The value of total dietary fiber found (18.35%) demonstrates the high functionality of Defatted Flour of Brazil Nuts. The levels of insoluble fibers found in this investigation (13.15%) were higher than those of soluble fibers (5.20%), constituting an important food source of fibers, as a basic raw material or as a product enrichment resource in the food industry [19]. Brazil nut flour is considered, according to the Brazilian legislation, as a product with high fiber content [19]. The recommendations of different international entities recommend a diet fiber intake in the range of 18 to 38 g per day [20].

The carbohydrate average content (5.40 g 100 g^−1^) was obtained by difference, and expresses the variations in the other components analyzed. This value is considered low when compared to the content found in sapucaia flour with averages of 13.76% to 34.65% [16] and in two varieties of peanut flours with averages of 18.17% to 20.04% [18]. Defatted Flour of Brazil Nuts water activity (0.25), moisture (4.45%) and fixed mineral residue (3.15%) were low due to the reduction in hydrophobic compounds most exposed to heat treatment (drying). These data show the stability of this flour and help predicting its richness in mineral elements, when compared with other types of flour [17].

This energetic-protein potential when used as a basic raw material for bar formulations provided a high nutritional and functional contribution to the elaborated product. When these energy value formulations ranging from 350.94 kcal 100 g^−1^ to 359.9 Kcal 100 g^−1^ kcal are related to a standard 2000 kcal diet, they represent a percentage of approximately 20% of the daily energy requirements.

A high concentration in the protein content was one of the findings in the elaborated product. However, the different concentrations showed no significant difference. The results were superior when compared to the research by Muniz et al. [4] with the incorporation of fermented by-products of guava and cashew bars (protein levels ranging from 10.74% to 12%). Similar results in comparative terms were found in the investigations by Sethupathy et al. [21] with the incorporation of granolas and other phospho-oligosaccharides (PO) in protein bars: the proteins average did not show significant difference (14.38% to 14.70%). 

In contrast, in the investigations by Costa et al. [22] with snacks formulations based on raw material with high protein content (81.43%), the final formulations showed protein variations of 12.52% to 19.51% varying increasing proportionally to the increase in protein-based raw material. These results are similar to what occurred in this investigation.

The concentration of protein components in bars is an attractive feature for the main target audience of these products, those who practice physical activities. This is due to the fact that the constitution and reconstitution of muscle tissue requires protein constituents for muscle hypertrophy (anabolism and muscle catabolism), as well as for the recovery of micro injuries caused by repetitive effort characteristic of the practice of physical activities. Protein content is of great importance for the food industry, especially sports food supplements manufacturers, because one of their largest market segments is represented by protein or high protein content bars, as compared to other kinds of bars (cereal bars, energy bars, fruit bars, fiber bars, diet bars, light bars, among others); in addition to being the bars with the greatest commercial value [1,4]. 

In this research the formulations of bars from by-products whose base raw material promotes a strong increase in the productive chain of the agricultural culture of Brazil nuts. In addition, all formulated bars had a protein content greater than twice that of commercial protein bars averaging 2–15 g 100 g^−1^.

Changes in the concentrations of the formulations base constituents, with a reduction in the presence of defatted nut flour and an increase in soy protein isolate, in the levels used in this research did not significantly alter the lipid content of the bars. This is explained in part by the reduced increase in soy protein isolate (20, 25 and 30%) and because both matrices are defatted.

The average lipid content of the formulated bars was the lowest among the energy macronutrients, with no significant difference between the formulations. When the bars are compared in the investigation by Muniz et al. [4] (value from 6.06% to 7.40%) the results were higher. On the other hand, in the research by Sethupathy et al. [21] (value from 16.33% to 16.71%) the lipid content was lower. In the investigations taken as a basis of comparison, the formulations did not show significant differences, in this constituent, a pattern similar to the present study.

The amount of total dietary fiber found in the formulations showed a significant difference only in formulation 3, which has the lowest proportion of Defatted Flour of Brazil Nuts. When compared to the formulations of granola bars Sethupathy et al. [21] also found differences in the crude fiber content in the formulations, with higher averages (13.70% to 22.42%) than those observed in this research.

However, all formulations with Defatted Flour of Brazil Nuts were considered a fiber source. According to the National Health Surveillance Agency (ANVISA), food products can be classified as a source of fiber or with high fiber content. The requirement for a product to be considered as a fiber source is that it should contain at least 3 g of fiber for every 100 g of total solids. In order to be considered with “high fiber content” the material must contain at least twice as much (6 g) [19].

The prominent presence of fibers in the formulations (Table 2) reveals the functionality expressed in the processed products, given the importance of this nutritional component in preventing diseases in pathologies associated with chronic non-communicable diseases and cardiovascular diseases [23].

In the formulations of the bars, the carbohydrate content did not show significant difference between the formulations, presenting the highest percentage value among the macronutrients, expressing the energy richness of the product. When comparing the formulations presented in the research by Sethupathy et al. [21] with average carbohydrates of 22.51% to 43.33%, results superior to formulations with increased sucrose and inulin were obtained. In formulations with fermented guava and cashew nuts used in bars [10] the carbohydrate averages were higher, varying from 48.85% to 54.46%.

These results are considered desirable, since one of the largest consumer market profiles for these products, appears in the segment of athletic performance [1,4]. The energetic wear caused by physical activity requires an increase in the total energy value (TEV) of those practicing this activity, in order to supply a metabolic demand greater than that recommended by the average daily reference values, for a 2000 kcal diet.

One of the most important parameters for product stability is water activity (Aa) and its moisture. The results (Aa close to 0.60) attribute good stability to bar formulations against possible changes caused by microorganisms and other chemical reactions [16,17,18]. Statistical data did not show significant differences between formulations a relevant fact in ready-to-eat snack products, as they infer important characteristics related to microbiological stability, packaging design, shelf life and reduced use of stabilizers.

The moisture content showed a significant difference in formulation 3, where the content of soy protein isolated (SPI) is higher, possibly because it is drier than Defatted Flour of Brazil Nuts. When compared to the bar formulations in the research by Sethupathy et al. [21] lower results in the formulations were found, with averages of 32.27% and 25.12% respectively. However, they were greater in the bar formulations in the investigation by Concenço et al. [24] with flaxseed flours (average of 9.7% to 10.3%).

The ash contents did not change, without significant differences at the level of 5%. These results, when compared to bar formulations with flaxseed flour, also did not show significant differences, with averages of 1.0% [24]. In the formulations by Sethupathy et al. [21] there was an average variation of 1.0% to 1.90%. Both investigations exhibited results inferior to those obtained in our investigation. This parameter is indicative of the mineral level in the mixture of DFBN and SPI as being the most prevalent constituent in those formulations.

### 3.2. Defatted Flour of Brazil Nuts and Bars Formulations Mineral Analysis

The mineral composition of Defatted Flour of Brazil Nuts and of the formulations are shown in Table 3; minerals were classified in macrominerals and microminerals. 

The mineral composition found in Defatted Flour of Brazil Nuts showed a high content in important macrominerals such as sodium, potassium, calcium and phosphorus; among the formulated bars, there were no differences at the level of 5% statistical significance for the minerals calcium, sodium and phosphorus. However, in formulation 3 there was a significant difference for the minerals potassium and magnesium. This fact is the result of the proportional addition of the soy protein isolate, which base, soy, is the source of these minerals.

The functionality related to the presence of minerals such as Ca, P and Na are in charge of the production of the electrical potential through the membranes by the Na/K/Ca/Atpase “pump” system, which are important in the regulation and maintenance of muscle stimuli across the cell membranes. Phosphorus has a great importance in the muscle phosphorylation processes, that is, the formation of the intramuscular substrate of great energy capacity, capable of resynthesizing adenosine triphosphate (ATP), providing immediate energy for muscle actions [26].

Among these macronutrients, the one with the greatest quantitative prominence is magnesium, which presented a value above the daily recommended intake for adults of 400 mg 100 g^−1^. However, the bars formulations in this investigation revealed only 30 g content, representing one third of the daily recommended intake. This mineral is considered important due to its performance in the preventive processes of cardiac arrhythmias, hydrodynamic blood regulation balance, inflammatory responses, among other functions [27].

Among the essential micro minerals presented in Table 4, iron, copper, manganese, zinc and selenium did not show statistical differences at 5% level between the different formulations. The presence of these micro minerals provides the functionality of this product as the rich minerals are directly associated to the enzymatic activities in synthesis reactions and/or degradation of energetic metabolites such as carbohydrates, lipids, proteins, nucleic acids, constituents that in general constitute an important part of the human body muscle content, among other functions such as those when in combination with Copper and Zinc acting in the production of mitochondrial energy, and also exerting an antioxidant protective action, promoting the synthesis of melanin and catecholamines [28]. 

The major constitution of Defatted Flour of Brazil Nuts and of the bar formulations, with high protein concentration and the amino acids cysteine and methionine favor the association of these with the micromineral selenium, developing primarily the selenocysteine and selenomethionine forms, adding to the copper, iron and manganese levels considered in the constitutions of protein with predominant fractions of albumin, as in the Brazil nut proteins. The content in the formulations did not show statistical significant differences at 5% level. These values represent approximately 50% of the requirement established by FAO [25] for adult individuals when related to the consumption of a 30 g bar formulation.

Selenium has been associated with vitamin E as an antioxidant element acting directly on the levels of the enzymes glutathione peroxidase, assisting in the modulation of the immune system, in thyroid hormones, with actions in the counting of auxiliary T cells, in the immune response to viruses in immunodeficient individuals, as an anti-free radical agent and coadjuvant in the prevention and treatment of different types of cancer [27,28].

These micronutrient composition data show the nutritional and functional importance of the elaborated products. Just as its importance in nutrition and use of broken nuts that do not fit the commercial standards; used in products with high commercial growth and practicality.

### 3.3. DFBN and Bar Formulations Amino Acids Profile 

The quality of a protein is assessed by its specific composition in amino acids and the ease with which it is digested and absorbed by the body. The total amino acids found in DFBN and in the bar formulations are shown in Table 4, compared to the reference standard for adults [10]. 

The values observed in the formulations show the amino acid Lysine as a limiting factor with a chemical score ranging from 0.21 to 0.23 with no difference at the 5% level of statistical significance. The reduced content of lysine in the formulations may be due to the presence of reducing sugars used in the preparations. An average reduction of 30% is observed in the level of Lysine in the formulations compared to the contents of the Defatted Flour of Brazil Nuts. Reducing sugars can lead to loss of lysine, a fact that may explain the low content of this amino acid (Table 5). Another fact is that under certain heat treatment conditions, proteins can become resistant to digestion, and consequently the bioavailability of some amino acids may be reduced [29].

The highest chemical scores were found in the sulfur-containing amino acids (methionine + cysteine) with a value of 1.43. The content of these amino acids shows a mg/g value close to that determined as a reference standard for adults aged 18 years and over (the group to which this product is targeted), which standard recommendation is 22 mg/g of protein [10]. Formulations F1 and F2 did not show significant differences in the methionine and cysteine content. However, the F3 formulation showed a significant difference at the level of 5% between the values. This result is related to the reduction of DFBN content in the F3 formulation from 30% to 20%, stressing that DFBN is rich in sulfur amino acids, and soy protein is deficient.

Another group of essential amino acids of great prominence in this product are the branched chain amino acids (BCAA), which correspond to isoleucine, leucine and valine, which showed values without significant differences in the results obtained in this study and with chemical scores average of 30% of the amino acid recommendations for adults [10]. These amino acids gain prominence, in this connection, due to their function in providing energy in extended physical activities, such as those of endurance (physical activities carried out for more than 90 min). 

The essential aromatic amino acids phenylalanine and tyrosine showed a significant difference in the formulation F3 where the content of isolated soy protein is at its highest level (30%) and soy is considered a source of these amino acids. The chemical score of these amino acids indicate that they meet more than 60% of the total amount recommended by FAO for adult individuals.

Regarding non-essential amino acids, the highest value found in the formulations was glutamine, without significant differences, with averages ranging from 46.67 to 47.2 mg g^−1^ of protein. This amino acid is one of the most important in the energy metabolism, its prompt and practical intake, like those of these products, can minimize the rates of microtrauma in skeletal muscles caused by over-training. The decrease in the availability of this amino acid for the cells of the immune system can cause a decrease in immunity, due to its connection with the induction of lymphocyte synthesis, making practitioners of physical activities susceptible to infectious processes.

Sensory Analysis—Acceptability tests applied to the formulations of the finished products allowed the acceptance indexes (AI) calculation for each formulation.

According to Cozzolino [29] the most significant consequences of changes caused by processing amino acids may be related to storage conditions, industrial processing, moderate heating and the presence of reducing sugars that can lead to loss of bioavailable lysine, a fact that may explain the low amount of this amino acid found in the bars formulations.

### 3.4. Acceptability Test 

The acceptance indexes (AI) obtained through the acceptance test (sensory analysis), performed with the different bar formulations are shown in Table 5. 

From comparative data, the F value for the color attribute, obtained, through the use of ANOVA, for the different bar formulations (Fcor = 2.34), with the tabulated F value (FT (5%) = 3.14 and FT (1%) = 4.95), it can be stated with 99% certainty, that there is no significant difference between the formulations for that attribute. Regarding the aroma attribute, an F value (Faroma = 10.67) was higher than the tabulated F value (FT (5%) = 3.14 and FT (1%) = 4.95). In this case, it can be stated with 99% confidence that there is a significant difference between the formulations for the aroma attribute. Applying the Tukey test, it was observed, with 95% confidence that the 75:25 formulation was the most accepted, considering the aroma attribute.

For the texture attribute, the F value (Ftexture = 6.35) was again higher than the tabulated F value (FT (5%) = 3.14 and FT (1%) = 4.95), indicating with 99% confidence that there is a significant difference between the formulations for the texture attribute. Applying the Tukey test, it was observed that the 75:25 formulation was also the most accepted regarding the texture attribute. The F value for the flavor attribute (Fsabor = 9.29), was higher than the tabulated F value (FT (5%) = 3.14 and FT (1%) = 4.95), which makes it possible to state with 99% confidence, that there is a significant difference for this attribute, for the evaluated product. Applying the Tukey test, it was observed that the 75:25 formulation was also the most accepted regarding the flavor attribute.

Comparing the F value for the general impression attribute (Fgeneral impression = 5.79) with the tabulated F value (FT (5%) = 3.14 and FT (1%) = 4.95), it can be said with 99% confidence, that this attribute showed a significant difference between the different bar formulations. Applying the Tukey test, it was observed, with 99% confidence, that the 75:25 formulation was the most accepted regarding the general impression aspect, based on the general results obtained with ANOVA and expressed in the average acceptance index in Figure 2. 

It is possible to verify with 95% statistical reliability that the evaluated bar type formulations present significant differences for the most objective attributes in the acceptability test (aroma, flavor, texture and overall impression of the product). The attribute that showed no significant difference was the color aspect; however, this shows more subjective results, which can offer more reliability when measured using analytical techniques.

From the general evaluations with statistical basis, the formulation of the F2 bar in the proportion 75:25 can be considered the most suitable for the scale of production and marketing, which gives it characteristics of a product with adequate flavor, aroma and texture and with good general acceptance.

### 3.5. Purchase Intention

The values obtained in the purchase intention test of the elaborated formulations are shown in Figure 3.

The charts show that the bars prepared with a concentration of 70:30 and 75:25 of defatted Brazil nut flour and isolated soy protein, can be products of great acceptance by the market.

In the analysis of the sum of the items scores labeled “would certainly buy” and “would probably buy”, the bars with 70:30 and 75:25 concentrations obtained the highest percentages (88.57% and 97.15%) of purchase intention, respectively. For formulations with an 80:20 concentration, this percentage was 68.57. The data presented confirm the market potential of this product.

The general evaluation obtained in the applied tests shows results of acceptability and purchase intention, which attributes technical standards to the product, (determined by its physical-chemical, microbiological, nutritional and functional properties) with great technological and commercial possibilities that could be developed in large scale by the food industry in this segment.

## 4. Conclusions

The data reflect all the energy potential of the Defatted Flour of Brazil Nuts, which increases the number of options in terms of regional raw materials that can be used by the food industry in the processing and/or enrichment of other food, or different types of flour, as a way to aggregate greater commercial value and energy-protein increase in food processing.

In the food industry, this source of protein that is concentrated in defatted Brazil nut flour can have a wide range of applications, based on protein quantity and quality, in addition to its functional properties evidenced by the fiber content, constitution of the acid profile unsaturated fatty acids present in the oil and the antioxidant action expressed primarily because it is one of the richest sources of selenium.

In this connection a Brazil nuts that do not comply with the export standards showed a high potential for industrial application. In addition to the extraction of their high lipid content, they produce a defatted by-product, with a high functional nutraceutical composition, expressed in their high content of proteins, fibers, macro and micro minerals and essential amino acids.

The combination of this nutraceutical potential with the isolated soy protein allowed to produce bars that maintained and expanded their organic functionality, carrying a high protein content, the functionality of soluble and insoluble fibers, the concentration of macro and micro minerals such as magnesium, calcium, sodium, potassium (active in the restructuring of the muscular-energetic system) with emphasis on zinc and selenium (with their recognized action on the immune and antioxidative systems). Added to the high chemical scores of amino acids (with emphasis on sulfuric, aromatic and branched chain amino acids) all present in the bar-type formulations.

The sensory and purchase intention analyses showed the preference of the tasters for the F2 formulation composed of 75% DFBN and 25% SPI with purchase intention close to 90% acceptability. Thus, the technical and commercial ratification of the feasibility of the development of bar-type products with Brazil nut by-products is ratified.

These data show the potential of using industrial by-products when these isolated or combined raw materials are applied in growing market segments, as strategies for new product designs with greater consumption and practicality combined with a composition of nutraceutical-functional elements for the human body. Finally, also helping generating expansion of the production chain, greater profitability and reduction of environmental impact. Since, in the northern region of Brazil, nut residues, in many cases, are discarded in the environment. Thus, the use of residues in the form of by-products of the nut can positively affect the environmental impact.

## Figures and Tables

**Figure 1 foods-12-01446-f001:**
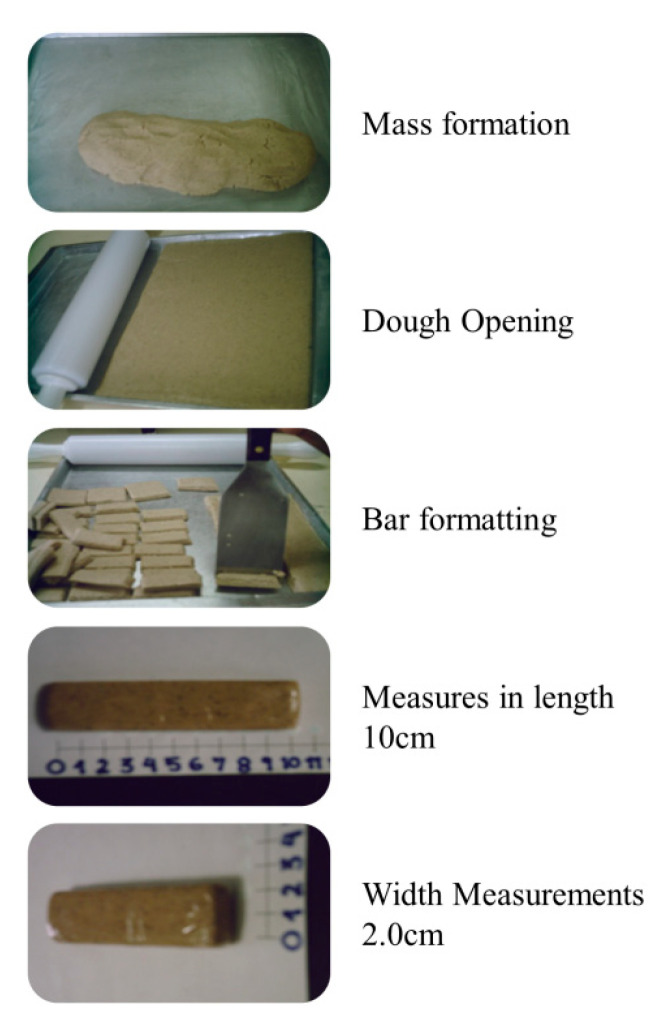
Product Elaboration Stages.

**Figure 2 foods-12-01446-f002:**
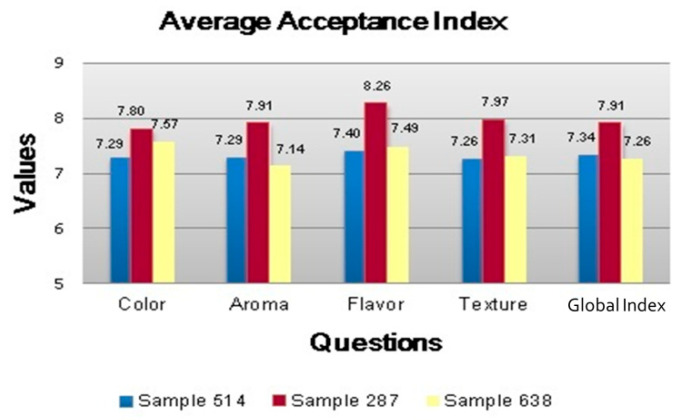
Average Acceptance Index.

**Figure 3 foods-12-01446-f003:**
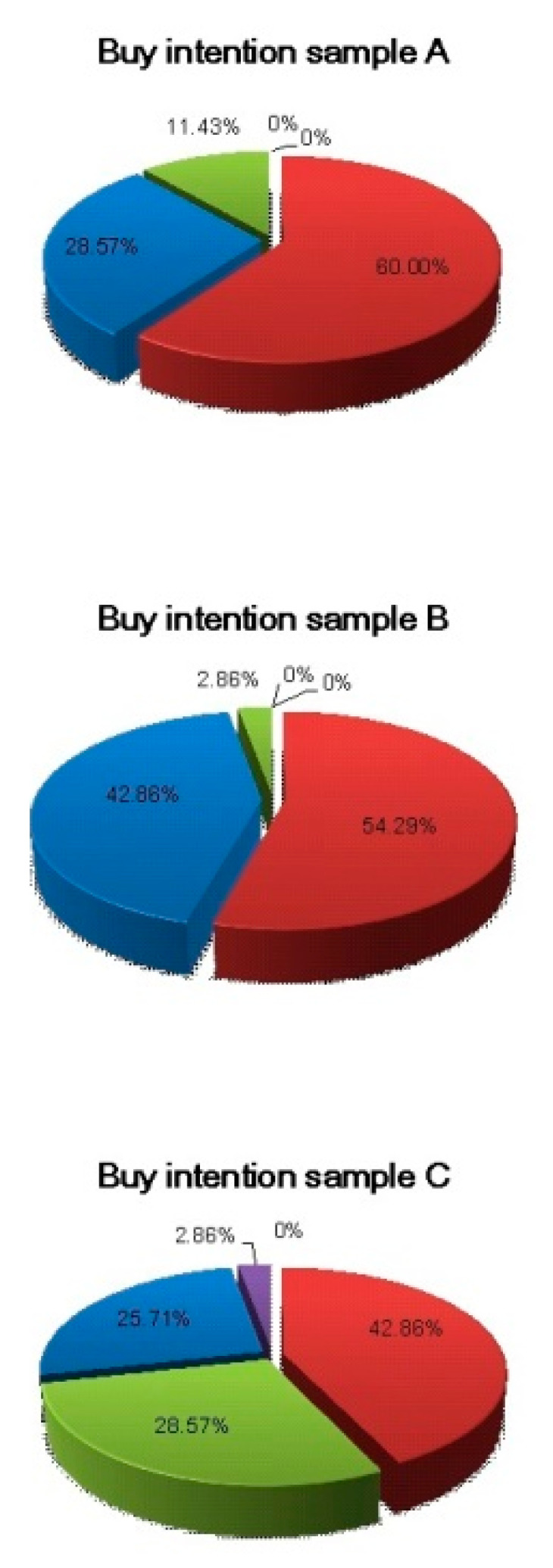
Purchase intention test.

**Table 1 foods-12-01446-t001:** Bar formulations made with Defatted Flour of Brazil Nuts (DFBN) and Soy Protein Isolate (SPI).

Ingredients (g)	Formulations		
	F1	F2	F3
	(DFBN 80: SPI20)	(DFBN 75: SPI 25)	(DFBN 70: SPI 30)
DFBN	240 g	225	210
SPI	60 g	75	90
binder syrup (mL)			
Inverted liquid sugar	105	105	105
Glucose syrup	75	75	75
Hydrogenated vegetable fat	15	15	15
Yield%	98.60	98.10	97.70

**Table 2 foods-12-01446-t002:** Physical and physical-chemical composition of Defatted Flour of Brazil Nuts and formulations.

Parameters	DFBN	Formulations		
		F1	F2	F3
Energy (kcal 100 g^−1^)	402.70	350.94	354.10	359.90
Water activity	0.25 ± 0.03	0.60 ± 0.02 ^a^	0.55 ± 0.05 ^a^	0.54 ± 0.02 ^a^
Carbohydrate (%)	5.40 ± 0.50	32.15 ± 0.45 ^a^	33.0 ± 0.02 ^a^	33.3 ± 0.52 ^a^
Protein (%)	47.35 ± 1.33	28.09 ± 0.32 ^a^	28.5 ± 1.02 ^a^	29.85 ± 0.75 ^a^
Lipids (%)	21.30 ± 0.85	12.22 ± 0.15 ^a^	12.54 ± 0.75 ^a^	11.93 ± 0.82 ^a^
Dietary fiber (%)	18.35 ± 1.03	9.85 ± 0.64 ^a^	9.03 ± 1.15 ^a^	8.90 ± 1.02 ^a^
Insoluble fiber (%)	13.15 ± 0.85	6.15 ± 0.15 ^a^	6.05 ± 0.83 ^a^	5.85 ± 0.72 ^a^
Soluble fiber (%)	5.20 ± 0.55	3.70 ± 0.55 ^a^	2.98 ± 0.50 ^a^	3.05 ± 0.02 ^a^
Umidity (%)	4.45 ± 0.50	15.10 ± 0.54 ^a^	14.55 ± 0.87 ^a^	13.85 ± 0.78 ^b^
Ash (%)	3.15 ± 0.23	2.59 ± 0.76 ^a^	2.35 ± 0.52 ^a^	2.15 ± 0.52 ^a^

When the means are statistically significantly different. (*p* > 0.005). Data expressed on a dry basis (mean of the triplicates ± standard deviation) F1—Formulation with 80% DFBN and 20% SPI; F2—Formulation with 75% DFBN and 25% SPI; F3—Formulation with 70% DFBN and 30% SPI.

**Table 3 foods-12-01446-t003:** Composition of macro and micro minerals in DFBN and the bars.

Minerals Macrominerals (mg/100g)					
Formulatiions					
	DFBN	F1	F2	F3	FAO * (2004) mg/Day
Sodium (Na)	3.96 ± 0.01	4.35 ± 0.55 ^a^	4.38 ± 0.35 ^a^	4.42 ± 0.27 ^a^	2400
Potassium (K)	610 ± 0.03	755 ± 0.75 ^a^	767 ± 0.75 ^a^	770 ± 0.75 ^b^	1000
Calcium (Ca)	44.80 ± 4.50	65.8 ± 11.50 ^a^	66.30 ± 11.50 ^a^	68.70 ± 11.50 ^a^	1000
Phosphor (P)	670 ± 10.50	652 ± 5.53 ^a^	654.3 ± 8.50 ^a^	658.20 ± 7.35 ^a^	1000
Magnesium (Mg)	725.80 ± 28.20	745 ± 15.5 ^a^	749 ± 15.50 ^a^	753 ± 15.50 ^b^	400
**Microminerals (mg/100g)**					
Iron (Fe)	5.08 ± 0.65	4.58 ± 0.15 ^a^	4.43 ± 0.15 ^a^	4.38 ± 0.15 ^a^	18
Copper (Cu)	4.01 ± 0.29	4.05 ± 0.73 ^a^	3.95 ± 0.73 ^a^	3.88 ± 0.73 ^a^	2
Manganese (Mn)	3.30 ± 0.39	3.96 ± 0.85 ^a^	3.89 ± 0.85 ^a^	3.82 ± 0.85 ^a^	2
Zinc (Zn)	4.51 ± 0.43	4.35 ± 0.35 ^a^	4.38 ± 0.34 ^a^	4.41 ± 0.56 ^a^	15
Selenium (Se) µg/g	1.11 ± 0.50	0.97 ± 0.43 ^a^	0.93 ± 0.30 ^a^	0.95 ± 0.53 ^a^	70 µg-men 55 µg-women

* Established requirement for adult FAO individuals [25]. When the means are statistically significantly different. (*p* > 0.005). DFBN—Brazil nut defatted flour, F1—Formulation with 80% DFBN and 20% SPI; F2—Formulation with 80% DFBN and 20% SPI; F3—Formulation with 70% DFBN and 30% SPI.

**Table 4 foods-12-01446-t004:** Total amino acid content (mg/g protein) of Defatted Flour of Brazil Nuts and bar formulations related to the reference standard [10] * and chemical scores.

Amino Acid	DFBN	EQ	F1	EQ	F2	EQ	F3	EQ	Standart FAO/WHO
**AA’S ESSENCIAIS**									
Histidine	8.00 ± 0.02	0.53	6.10 ± 0.03 ^a^	0.40	5.73 ± 0.21 ^a^	0.38	5.91 ± 0.03 ^a^	0.39	15
Threonine	9.20 ± 0.04	0.40	7.60 ± 0.02 ^a^	0.33	6.8 ± 0.02 ^a^	0.29	6.91 ± 0.02 ^a^	0.30	23
Lysine	13.20 ± 0.12	0.22	10.4 ± 0.02 ^a^	0.23	10.1 ± 0.02 ^a^	0.22	9.85 ± 0.05 ^a^	0.21	45
Methionine **	22.50 ± 0.21	1.40	16.4 ± 0.10 ^a^	1.02	15.8 ± 0.10 ^a^	0.98	15.0 ± 0.15 ^b^	0.93	16
Cysteine	5.50 ± 0.21	0.91	3.2 ± 0.10 ^a^	0.53	2.92 ± 0.10 ^a^	0.48	2.80 ± 0.11 ^b^	0.46	6
Isoleucine	12.20 ± 0.10	0.40	9.6 ± 0.08 ^a^	0.32	9.1 ± 0.08 ^a^	0.30	9.0 ± 0.18 ^a^	0.30	30
Leucine	24.70 ± 0.13	0.41	19.1 ± 0.09 ^a^	0.32	18.5 ± 0.09 ^a^	0.31	18.61 ± 0.10 ^a^	0.32	59
Valine	15.00 ± 0.12	0.38	11.5 ± 0.08 ^a^	0.29	10.7 ± 0.08 ^a^	0.27	10.65 ± 0.11 ^a^	0.27	39
Phenylalanine ***	14.55 ± 0.10	0.48	15.6 ± 0.10 ^a^	0.66	16.2 ± 0.10 ^a^	0.68	17.8 ± 0.12 ^b^	0.75	38
Tyrosine	8.95 ± 0.10		9.7 ± 0.07 ^a^		9.98 ± 0.07 ^a^		10.97 ± 0.05 ^b^		
Tryptophan	3.85 ± 0.08	0.64	ND	ND	ND		ND		6
Total	137.65		109.20		105.83		107.50		277
**Non-Essential Amino Acids**									
Asparagine	27.5 ± 0.15	-	24.1 ± 0.18 ^a^	-	24.34 ± 0.03 ^a^	-	24.61 ± 0.52 ^a^	-	-
Glutamine	72.5 ± 0.21	-	47.2 ± 0.12 ^a^	-	46.72 ± 0.52 ^a^	-	46.67 ± 0.12 ^a^	-	-
Serine	15.7 ± 0.30	-	12.0 ± 0.21 ^a^	-	12.12 ± 0.21 ^a^	-	12.15 ± 0.21 ^a^	-	-
Glycine	16.7 ± 0.87	-	11.1 ± 0.10 ^a^	-	10.71 ± 0.70 ^a^	-	10.67 ± 0.53	-	-
Arginine	25.3 ± 1.02	-	23.0 ± 0.14 ^a^	-	22.7 ± 0.47 ^a^	-	22.93 ± 0.38 ^a^	-	-
Alanine	13.5 ± 0.91	-	9.8 ± 0.05 ^a^	-	8.92 ± 0.13 ^a^	-	8.97 ± 0.35 ^a^	-	-
Proline	16.3 ± 0.21	-	11.9 ± 0.07 ^a^	-	11.76 ± 0.07 ^a^	-	11.07 ± 0.07 ^a^	-	-
Total (mg/g)	325.15		248.3		243.10		244.57		277

* Amino acid content recommended for adults over the age of 18 years [10], Methionine + Cysteine **. Phenylalanine + Tyrosine ***. When the means are statistically significantly different (*p* < 0.005). F1—Formulation with 80% DFBN and 20% SPI; F2—Formulation with 75% DFBN and 25% SPI; F3—Formulation with 70% DFBN and 30% SPI. ND—not detected.

**Table 5 foods-12-01446-t005:** Acceptance index (AI) for the different formulations of the bars.

Formulações	AI (Color)	AC (Aroma)	AC (Flavor)	AC (Texture)	AC (Impressão Global)
F1	81.00 ^a^	80.0 ^a^	82.22 ^a^	80.67 ^a^	81.56 ^a^
F2	86.67 ^a^	87.89 ^b^	91.78 ^b^	88.56 ^b^	87.89 ^b^
F3	84.11 ^a^	79.33 ^a^	83.21 ^a^	81.22 ^a^	80.67 ^a^

Means with the same letters, in the same column, do not differ statistically (*p* > 0.005). F1-Formulation with 80% DFBN and 20% SPI; F2- Formulation with 75% DFBN and 25% SPI; F3-Formulation with 70% DFBN and 30% SPI.

## Data Availability

The data used to support the findings of this study can be made available by the corresponding author upon request.

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
