# Peer review of "Development of a Nutraceutical Product Derived from By-Products of the Lipid Extraction of the Brazil Nut (Bertolletia excelsa H.B.K)"

_foods, 2023, doi:10.3390/foods12071446_

Round 1
Reviewer 1 Report
Authors well utilised the brazil nut extract for preparation of the bars. I really appreciate the thoughts and efforts made in formulating with all proper measurements and regulations.
However I have some concerns about this work. 1) Authors did not discuss the shelf life of the bar. Was this factor studied? 2) Did the authors perform accelerated stability studies of the bar? This is considered as an important factor as they will be transported across the countries which might affect the composition of the bar. 3) Did authors identify any levels of inorganic compounds such as heavy metals like lead etc or other contaminants during analysis? 4) Authors did not discuss the preservatives and colorants used in the formulation. 5) There is mere information in the figure and table legends. Kindly eloborate
Author Response
Reviewer Questions
1) Authors did not discuss the shelf life of the bar. Was this factor studied?
I appreciate the question because the shelf life of the bar has not yet been studied.
2) Did the authors perform accelerated stability studies of the bar? This is considered as an important factor as they will be transported across the countries which might affect the composition of the bar.
I appreciate the question, because the accelerated stability of the bar has not yet been realized.
3) Did authors identify any levels of inorganic compounds such as heavy metals like lead etc or other contaminants during analysis?
Heavy metals were not evaluated. In relation to other contaminants by microbiological and mycotoxicological analyzes, contaminants were not detected.
.4) Authors did not discuss the preservatives and colorants used in the formulation. 5) There is mere information in the figure and table legends. Kindly eloborate.
Neither preservatives or dyes in the elaboration of the product were used.
Reviewer 2 Report
REVIEW
For the journal Foods (ISSN 2304-8158)
Article “Development of a nutraceutical product derived from by-products of the lipid extraction of the Brazil nut (Bertolletia excelsa H.B.K)”
Manuscript ID: foods-2240067
Authors: Orquidea Vasconcelos Santos, Glaucinéia Azevedo, Ângela Santos, Alessandra Santos Lopes
1. Previous research has shown that industrial Brazil nuts by-products have high nutritional importance, due to high levels of nutritional and functional constituents in these raw materials. The aim of this study was to develop a nutraceutical bar product using defatted Brazil nut by-products. The insertion of these nuts in the segment of products with high practicality as in bars, enhances the profile of consumers seeking daily food based on the tripod nutrition, prevention and consumption convenience, therefore the topic discussed by the authors is relevant both from a theoretical and practical point of view.
2. The authors should improve the quality and clarity of the notes in Figure 1 and clarify the title: "- Processus de formulation" (line 110).
3. Lines 111-112: “2.2.4. Microbiological and Physicochemical Analysis of the Defatted Flour of Brazil Nuts and Bar 111 Formulations”. Other similar sections are not italicized?
4. The statistical analysis section is imprecise and inconsistent.
What software and methods of statistical analysis were used, what statistical indicators were calculated for the analysis of physical-chemical and sensory tests?
It is not clear whether the authors themselves performed the comparison according to the table they cited ("The ANOVA F values were compared with tabulated F values (FT), provided by Dutcosky [13]" lines 152-153) or they only used the results calculated by the software?
5. The authors need to clarify the incomplete sentence: "Tukey's test (p≤0.05)." (Line 154).
6. Line 184. "Means with the same letters, in the same column, do not differ statistically (p> 0.005)". Instead of this sentence, I suggest that authors write when the means are statistically significantly different.
7. The same note for the statement in lines 307-308 and line 420.
8. Line 528. The year should be highlighted in the bibliographic description of the first literary reference.
9. The article is interesting, but the adjustments mentioned are recommended.
Sincerely, reviewer.
Author Response
Reviewer Questions
- The authors should improve the quality and clarity of the notes in Figure 1 and clarify the title: "- Processus de formulation" (line 110).
The title of Figure 1 has been changed to: Product Elaboration Stages.
- Lines 111-112: “2.2.4. Microbiological and Physicochemical Analysis of the Defatted Flour of Brazil Nuts and Bar 111 Formulations”. Other similar sections are not italicized?
This was done.
- The statistical analysis section is imprecise and inconsistent. What software and methods of statistical analysis were used, what statistical indicators were calculated for the analysis of physical-chemical and sensory tests?
This was done.
It is not clear whether the authors themselves performed the comparison according to the table they cited ("The ANOVA F values were compared with tabulated F values (FT), provided by Dutcosky [13]" lines 152-153) or they only used the results calculated by the software?
The results calculated by the software were compared to those tabulated.
- The authors need to clarify the incomplete sentence: "Tukey's test (p≤0.05)." (Line 154).
This was done.
- Line 184. "Means with the same letters, in the same column, do not differ statistically (p> 0.005)". Instead of this sentence, I suggest that authors write when the means are statistically significantly different.
This was done.
- The same note for the statement in lines 307-308 and line 420.
This was done.
- Line 528. The year should be highlighted in the bibliographic description of the first literary reference.
This was done.
Reviewer 3 Report
Dear authors, attached are my few suggestions for improvement of the article.

Author Response
Reviewer Questions
Figure 1 the photos presented in Figure 1 are not clear. The images are inapposite, it is necessary to redo Figure 1.
This was done.
Figure 2 What does GI mean? Global impression? If it is, the authors must explain in the legend.
This was done.
Pag.13 line 510: in the conclusions, the authors also indicate the reduction of the environmental impact. They should explain better why.
This was done.